# Adsorption of Mixed Dispersions of Silica Nanoparticles and an Amphiphilic Triblock Copolymer at the Water–Vapor Interface

**Carlo Carbone [1], Alejandra Rubio-Bueno [1], Francisco Ortega [1,2], Ramón G. Rubio [1,2] and Eduardo Guzmán [1,2,*]**

[1] Departamento de Química-Física, Facultad de Ciencias Químicas, Universidad Complutense de Madrid, Ciudad Universitaria s/n, 28040 Madrid, Spain; carlcarb@ucm.es (C.C.); alejru07@ucm.es (A.R.-B.); fortega@quim.ucm.es (F.O.); rgrubio@quim.ucm.es (R.G.R.)

[2] Instituto Pluridisciplinar, Universidad Complutense de Madrid, Paseo Juan XXIII 1, 28040 Madrid, Spain

\* Correspondence: eduardogs@quim.ucm.es; Tel.: +34-91-394-4107

**Abstract:** This study investigates the surface modification of hydrophilic silica nanoparticles by non-chemical adsorption of an amphiphilic triblock copolymer, Pluronic F-127, and elucidates its influence on the interfacial dispersion properties. The interaction between Pluronic F-127 and silica nanoparticles drives the formation of copolymer-decorated particles with increased hydrodynamic diameter and reduced effective charge as the copolymer concentration increases, while the opposite effect occurs as the particle concentration increases at a fixed polymer concentration. This indicates that increasing the copolymer concentration leads to an increase in the coating density, whereas increasing the particle concentration leads to a decrease. This is of paramount importance for modulating the reorganization of the Pluronic F-127 shell upon adsorption at fluid–fluid interfaces and, thus, the adsorption of the decorated nanoparticles at the interface and the rheological properties of the obtained layers. In fact, the relationship between copolymer concentration and interfacial tension, as well as the mechanical response of the interface, mirrors the patterns observed in Pluronic F-127 solutions, and only a shift mediated by the Pluronic F-127 concentration is found. This suggests that the presence of particles limits the space available for Pluronic F-127 molecules to reorganize at the interface but does not significantly affect the interfacial behavior of the particle-laden interface.

**Keywords:** adsorption; amphiphilic triblock copolymers; association; complexes; dilational rheology; fluid interfaces; hydrogen bonds; silica nanoparticles



## 1. Introduction

The stability of colloidal dispersions can be tuned by physisorption of surfactants or polymers onto their surface through soft interactions such as electrostatic interactions, hydrogen bonding, or van der Waals forces [1–3]. The occurrence of such associations results in the formation of complexes that have different properties from the original particles. In fact, the formation of these complexes can modify the interparticle interactions and their ability to adsorb at fluid–fluid interfaces [4,5], which are very important issues for controlling the stability of thin films, emulsions, and foams [6,7]. For example, silica nanoparticles are often extremely hydrophilic and tend to remain dispersed in the bulk, which limits their use in stabilizing emulsions or foams. However, the addition of molecules that can interact with their surface can help to improve their ability to adsorb at fluid–fluid interfaces [8,9]. This has been demonstrated by several authors by using surfactants as modifiers, most of them interacting through electrostatic interactions [10–13] or directly grafted through chemical bonds [14,15].

The enhanced adsorption capacity of modified nanoparticles at fluid–fluid interfaces, in contrast to bare particles, can be ascribed to the surface-active nature of the additives [8]. This effect is particularly noticeable when polymer-capped nanoparticles are used, as

they are able to form highly stable emulsions, whereas both unmodified particles and the capping polymer do not exhibit suitable emulsifying properties in solution [14,16]. However, the specific behavior of the polymer responsible for this enhanced efficacy is still poorly understood. Unlike unmodified nanoparticles, which have a limited capability to significantly reduce interfacial tension (at liquid–liquid interfaces) or surface tension (at liquid–vapor interfaces) through spontaneous adsorption from low-concentration suspensions, capped nanoparticles have the capacity to lower surface tension due to the presence of surface-active molecules bound to their surfaces. This drives the formation of a particle shell at the interface [14], which is relevant to various scientific fields, including physics, chemistry, and materials science; therefore, the motivation to study particle adsorption particles at fluid–fluid interfaces is multifaceted. First, interfacial adsorption plays a central role in several natural and industrial scenarios. In biological systems, the adsorption of proteins and surfactants on cell membranes affects cell function and communication. In industrial applications such as emulsions and foams, adsorption of particles at liquid interfaces affects stability, rheology, and overall product quality. In addition, the manipulation of interfacial adsorption has practical implications. Tailoring the adsorption of nanoparticles or molecules can lead to advanced materials with tunable properties, such as enhanced catalytic activity, improved drug delivery systems, and innovative sensors. This knowledge has the potential to revolutionize many industries, from healthcare to energy [16].

This research aims to assess the effect of Pluronic F-127, a triblock copolymer composed of two poly(ethylene oxide) terminal segments and a central poly(propylene oxide) segment, on the modification of the adsorption capabilities of hydrophilic silica nanoparticles at water–vapor interfaces, as well as on the interfacial rheology of the formed layers. Several studies have shown that the presence of ether groups in the poly(ethylene oxide) blocks of Pluronic polymers favors their association with hydrophilic silica nanoparticles through hydrogen bonding [17–19], allowing for the preparation of decorated particles using a single-step adsorption methodology [20]. This provides an intriguing opportunity to exploit the physicochemical properties of silica nanoparticles, which have limited affinity for fluid–fluid interfaces, to manipulate and enhance the properties of such interfaces.

## 2. Material and Methods

### 2.1. Chemicals

A triblock copolymer of the Pluronic family, Pluronic F-127, supplied by Merck KGaA (Darmstadt, Germany), was used in this work. Pluronic F-127 consists of two lateral blocks of poly(ethylene oxide) and a central block of poly(propylene oxide), with molecular weights of approximately 4.4 kDa and 3.8 kDa for poly(ethylene oxide) and poly(propylene oxide), respectively. The silica nanoparticles used were Ludox® HS-40 colloidal silica, provided as an aqueous dispersion containing 40% *w/w* solids, also supplied by Merck KGaA (Darmstadt, Germany).

Milli-Q-grade ultrapure deionized water (resistivity $\geq 18$ MΩ·cm, total organic content (TOC) < 6 ppm) was used to clean all materials and to prepare solutions and dispersions. This water was obtained using an AquaMAX™-Ultra 370 Series multi-cartridge purification system (Young Lin Instrument Co., Ltd., Gyeonggi-do, Republic of Korea).

### 2.2. Preparation of Pluronic F-127 Solutions and Pluronic F-127-Silica Nanoparticle Mixtures

Solutions and dispersions were prepared by weight using a precision analytical balance with an accuracy of $\pm 0.1$ mg. For the Pluronic F-127 solutions, the required amount of solid Pluronic F-127 needed to obtain solutions with concentrations ranging from 0 to 10 mg/mL was weighed, poured into a flask, and then solubilized with water to obtain the desired solution composition. For the dispersions containing Pluronic F-127 and silica nanoparticles, the first step was to weigh the appropriate amount of Pluronic F-127 required to obtain mixtures with copolymer concentrations ranging from 0 to 10 mg/mL. The weighed copolymer was then poured into a flask, and the amount of particles required to produce mixed dispersions with two different concentrations of silica nanoparticles

(0.1% $w/w$ and 1% $w/w$) was combined with the Pluronic F-127 in the same flask. Finally, water was added up to the final composition of the resulting mixtures is reached. It should be noted that the solutions and mixed dispersions were kept under stirring at 1000 rpm overnight to ensure their homogeneity. The pH of all dispersions and solutions was in the range 6.2–6.7, as determined by means of a pH meter (model CG842, Schott GmbH, Columbus, OH, USA) fitted with a Blueline 18 pH electrode (SI Analytics, Mainz, Germany).

### 2.3. Techniques

#### 2.3.1. Characterization of Bulk Dispersions

Dynamic light scattering (DLS) experiments were performed in a quasi-backscattering configuration, with a scattering angle ($\theta$) of 173°, using a Zetasizer Nano ZS instrument from Malvern Instruments Ltd. (Malvern, UK). The DLS experiments were performed using red-line radiation emitted by a He–Ne laser at a wavelength ($\lambda$) of 632 nm [21]. The DLS experiments allowed us to obtain the apparent diffusion coefficient, $D_{app}$, at a constant temperature of 22 °C for scatters dispersed in a liquid, assuming their Brownian motion. From the values of $D_{app}$, it is possible to evaluate the size of the scatters in terms of the apparent hydrodynamic diameter of the scatters, $d_h^{app}$, assuming the validity of the Stokes–Einstein equation:

$$d_h^{app} = \frac{k_B T}{3\pi\eta D_{app}}, \tag{1}$$

where $k_B$ and $T$ are the Boltzmann constant and the absolute temperature, respectively, and $\eta$ is the viscosity of the continuous phase. It should be noted that the DLS technique can only be used to analyze transparent dispersions in such a way that multiple scattering phenomena can be avoided.

The effective charge density of Pluronic-F-127-decorated silica nanoparticles can be determined by measuring the electrophoretic mobility ($u_e$) using laser Doppler velocimetry with a Zetasizer Nano ZS instrument from Malvern Instruments Ltd. (Malvern, UK). The electrophoretic mobility is directly proportional to the zeta potential ($\zeta$), which provides a quantification of the effective charge carried by the colloids dispersed in the aqueous medium [22,23].

#### 2.3.2. Study of the Adsorption at the Water–Vapor Interface

a.　Surface tension measurements

The dependence of surface tension on the Pluronic F-127 concentration was determined for both the F-127 solutions and the dispersions of Pluronic-F-127-decorated silica nanoparticles. This was carried out using a PS4 surface force tensiometer from Nima Technology (Coventry, UK) equipped with disposable paper Wilhelmy plates (Whatman CHR1 chromatography paper, Whatman, Maidstone, UK). The evolution of the surface tension of the water–vapor interface was measured until equilibrium was reached, i.e., changes in surface tension did not exceed 0.1 mN-m$^{-1}$ during a period of 30 min. Precautions were taken to minimize the influence of evaporation during the measurements. The data reported for each experiment represent the average of three independent measurements. All experiments were carried out at a constant temperature of 22.0 ± 0.1 °C. A more detailed description of the experimental procedure can be found in our previous publication [24].

b.　Dilational rheology

Oscillatory barrier experiments [25] were carried out using a NIMA 702 Langmuir balance from Nima Technology (Coventry, UK) equipped with a surface force tensiometer (PS4, Nima Technology, Coventry, UK). This setup allows the time evolution of the surface tension response to sinusoidal changes in surface area to be measured. This makes it possible to determine the dilational viscoelastic moduli of the interfacial layers, defined as $\varepsilon^* = \varepsilon' + \mathrm{i}\,\varepsilon''$ (where $\varepsilon'$ is the dilational elastic modulus and $\varepsilon''$ is the viscous modulus), over a frequency range of $10^{-1}$ to $10^{-2}$ Hz, and at a fixed surface deformation amplitude $\Delta u = 0.1$. The chosen deformation amplitude was checked to be appropriate to ensure that

the results obtained were within the linear response regime of the interface. It should be noted that the imaginary component of the dynamic surface elasticity ($\varepsilon''$) remained below 5% of the real component ($\varepsilon'$) in all measurements. Consequently, the detailed discussion of the imaginary part provides limited insight for the clarity of the paper and is therefore not discussed here.

## 3. Results

### 3.1. Characterization of Bare Silica Nanoparticles and Pluronic F-127

Before studying the association between Pluronic F-127 and silica nanoparticles in an aqueous medium, it is important to evaluate the behavior of the unmodified silica nanoparticles and Pluronic F-127 in an aqueous medium. The silica nanoparticles used had an average hydrodynamic diameter of about $15 \pm 4$ nm, as determined by DLS, and an average $\zeta$ potential of ($-35.7 \pm 0.8$ mV), which is consistent with the results previously reported by Liu et al. [26], indicating a negative charge due to the presence of dissociated silanol groups on the nanoparticles' surface. This is of paramount importance to ensure the stability of the nanoparticles in an aqueous medium. Despite the effective negative charge of the particles, this does not imply complete dissociation of all of the silanol groups on the surface of the nanoparticles; therefore, it is possible that hydrogen bonding interactions may occur through non-dissociated silanol groups on the surface of the nanoparticles.

The characterization of Pluronic F-127 aqueous solutions has been extensively discussed in our previous publication [17], and only the most fundamental aspects will be highlighted here. As the weight fraction of Pluronic F-127 in the solution increases, the viscosity of the solution increases progressively. This increase in viscosity is particularly noticeable when the copolymer weight fraction reaches about 120 mg/mL, with a sol–gel transition occurring at a copolymer weight fraction of about 200 mg/mL *w/w*. This sol–gel behavior is consistent with findings from previous studies on various polymers within the Pluronic family [27–30]. This transition limits the range of Pluronic F-127 concentrations that can be effectively used, and we therefore limited our study to a maximum concentration of 10 mg/mL, where the viscosity of Pluronic F-127 solutions and their mixtures with silica remains close to that of Milli-Q water.

### 3.2. Characterization of Dispersions of Pluronic-F-127-Decorated Silica Nanoparticles

The interaction of silica nanoparticles with Pluronic F-127 leads to the formation of copolymer-decorated silica nanoparticles, as demonstrated in our previous work [17] for the combination of Pluronic F-127 and another type of silica nanoparticles (Ludox® TMA). This is consistent with the results reported by Sarkar et al. [18], who proposed that copolymers belonging to the Pluronic family can adsorb on the silica surface regardless of whether it is protonated or not, forming a shell that has different conformations depending on the Pluronic concentration considered.

The association between Pluronic F-127 and silica nanoparticles can be rationalized considering the formation of hydrogen bonds between the ether groups present in Pluronic F-127 and the non-dissociated silanol groups on the surface of the silica nanoparticles. It has previously been reported that the formation of hydrogen bonds between the surface of silica nanoparticles and various polymers is a good alternative to enhance the stability of silica nanoparticle dispersions [31]. In fact, these hydrogen bonds are significantly stronger (about 25–30%) than the interactions between silanol groups and water molecules, favoring the association between Pluronic F-127 and the silica nanoparticles [32]. Consequently, when Pluronic F-127 is introduced into the colloidal dispersion, water molecules on the nanoparticle surface are replaced by copolymer molecules, resulting in the formation of copolymer-decorated silica nanoparticles. It is expected that increasing the number of available silanol groups on the surface of silica nanoparticles by mild acidification of the dispersions may improve the efficiency of the silica–Pluronic F-127 association. However, lowering the pH of the dispersion may increase the aggregation of the silica

nanoparticles, which will affect both the bulk association and the interfacial properties of the obtained dispersions.

DLS measurements provide empirical support for the interaction between copolymer and silica nanoparticles. As an illustrative example, Figure 1 shows a series of intensity autocorrelation functions ($g^{(2)}(t)$-1) and the resulting distributions of apparent hydrodynamic diameters obtained from their analysis. These results were derived from measurements of mixed dispersions consisting of silica nanoparticles and Pluronic F-127, where the concentration of the copolymer was progressively increased.

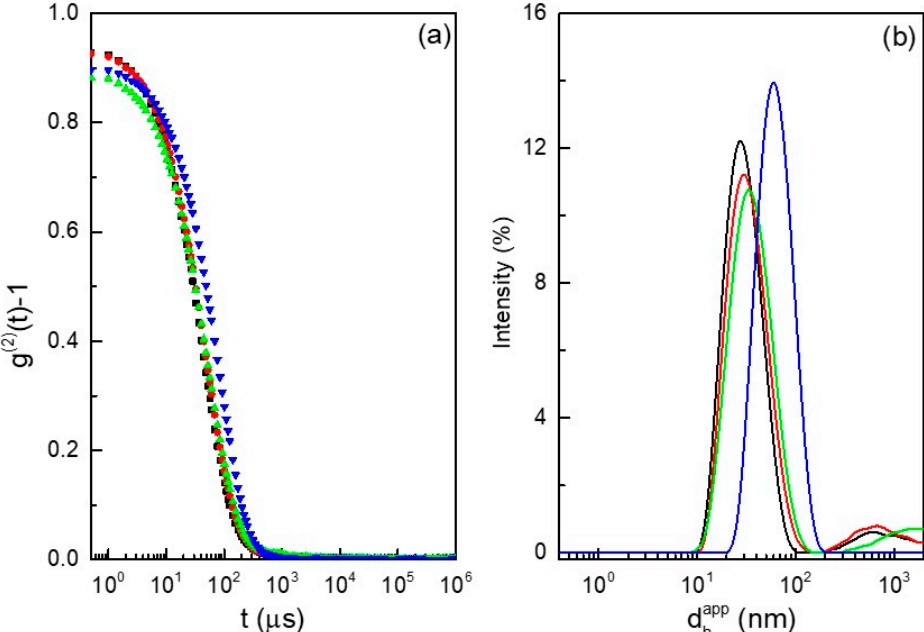

**Figure 1.** DLS results for mixed dispersions of silica nanoparticles (concentration, 1%$w/w$) and Pluronic F-127, with increasing concentrations of the latter: (**a**) intensity autocorrelation functions; (**b**) intensity–apparent hydrodynamic diameter distributions obtained from the analysis of the intensity autocorrelation functions in (**a**). The color code is the same in both panels (concentrations refer to Pluronic F-127): (■, —) 0.2 mg/mL, (●, —) 1 mg/mL g/L, (▲, —) 3 mg/mL, and (▼, —) 10 mg/mL.

The intensity autocorrelation functions obtained from the DLS experiments show a monomodal character regardless of the amount of Pluronic F-127 present in the colloidal dispersions. This indicates the presence of a single population of scatters within the dispersion, as supported by the intensity–apparent hydrodynamic diameter distributions shown in Figure 1b. However, it should be noted that the population appearing at the highest values of apparent hydrodynamic diameter must be considered meaningless, appearing as an artifact in the analysis of the intensity autocorrelation function due to the presence of a reduced number of aggregates or dust particles in the dispersion. It should be noted that the scattered intensity increases by a factor of $10^6$ with the characteristic dimensions of the scatters. Therefore, it is reasonable to expect that the scatter intensity for a significant scatter population in the region of high apparent hydrodynamic diameter would be significantly higher than that of the population observed at lower apparent hydrodynamic diameter values [21,33,34].

A more detailed analysis of the DLS results indicates that the characteristic relaxation time of Pluronic-F-127-decorated silica nanoparticles increases progressively with increasing Pluronic F-127 concentration (see Figure 1a), which can be interpreted as an increase in the characteristic size of the colloidal particles, evaluated in terms of the apparent hydrodynamic diameter, due to the formation of the copolymer layer on their surface (see Figure 1b). Therefore, based on the DLS results, a very efficient adsorption of Pluronic F-127 on the surface of silica nanoparticles can be assumed, which is consistent with previous results

reported by Sánchez-Arribas et al. [17]. It should be noted that average size obtained by DLS for decorated silica nanoparticles is higher than that expected for Pluronic F-127 chains, and that in the case of Pluronic F-127 solutions above the cmc, DLS shows a multimodal system where two different types of species (i.e., single chains and micelles) are involved, leading to more complex DLS results [17]. For a deeper understanding of the formation of Pluronic-F-127-decorated silica nanoparticles, Figure 2 shows the dependence on the average apparent hydrodynamic diameter (obtained as the maximum of the intensity–apparent hydrodynamic diameter distribution) on the concentration of the Pluronic F-127 included in the dispersion ($c_{\text{Pluronic F-127}}$) for dispersions with two different concentrations of silica nanoparticles ($c_{\text{NP}}$). It should be noted that the reported apparent hydrodynamic diameter is a concentration-dependent quantity, with this dependence resulting from interparticle interactions, and is therefore different from the true hydrodynamic diameter reported in Section 3.1 [35,36].

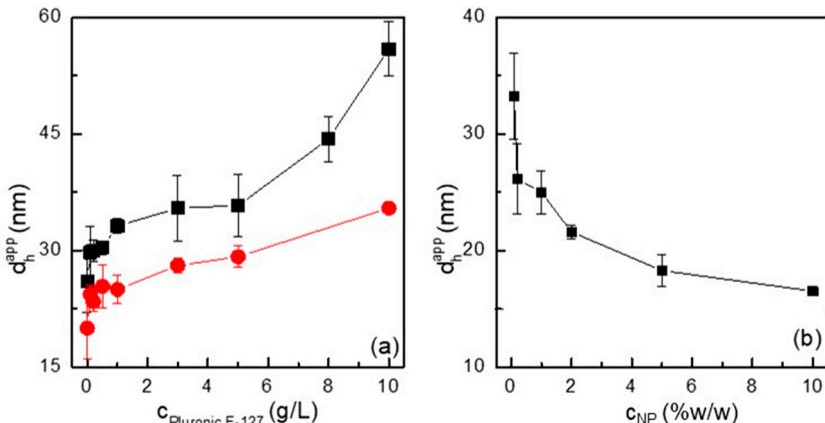

**Figure 2.** (**a**) Dependence of the average apparent hydrodynamic diameter on the concentration of Pluronic F-127 for the mixed dispersions. (■) $c_{\text{NP}}$ = 0.1% *w/w* and (•) $c_{\text{NP}}$ = 1% *w/w*. (**b**) Dependence of the average apparent hydrodynamic diameter on the nanoparticle concentration for silica nanoparticle dispersions with Pluronic F-127, where the concentration of Pluronic F-127 is fixed at 1 mg/mL. The lines in both panels are guides for the eyes.

The addition of Pluronic F-127 to the silica nanoparticle dispersions results in a significant increase in the apparent hydrodynamic diameter of the nanoparticles when dispersions with a fixed nanoparticle concentration are considered. In fact, the apparent hydrodynamic diameter ($d_h^{app}$) increases from approximately 20 nm for the unmodified particles to roughly 60 nm for dispersions containing a 0.1% *w/w* nanoparticle concentration, and up to values of about 30 nm when the concentration of the dispersions is increased by a factor of 10 (1% *w/w*). These observations strongly suggest the adsorption of Pluronic F-127 molecules onto the silica nanoparticle surfaces. At lower Pluronic F-127 concentrations, the copolymer molecules are likely to be adsorbed individually, adopting a flat conformation that becomes increasingly disordered with increasing copolymer concentration, as a result of an increase in the number of polymer segments protruding into the aqueous phase—and the consequent increase in $d_h^{app}$. However, once the critical micelle concentration (cmc) of Pluronic F-127 is overcome, micelles of the copolymer are expected to adsorb directly onto the nanoparticle surfaces. These micelles can be deformed as their number increases to maximize their adsorption on the surface. This is particularly important at the lowest nanoparticle concentration, where the dimensions of the Pluronic F-127 capping layer exceed the average apparent hydrodynamic diameter of the Pluronic F-127 micelles (approximately 15–20 nm) [17]. The adsorption of Pluronic F-127 micelles on silica surfaces is consistent with previous reports suggesting that the micellization of Pluronic copolymers on hydrophilic surfaces is preferential compared to the micellization process occurring in bulk aqueous phases [37,38].

Exploring the effect of the nanoparticle concentration on the adsorption of Pluronic F-127 onto the silica surface in more detail, it is important to note that the apparent thickness of the Pluronic F-127 layer increases as the concentration of nanoparticles is decreased. This can be explained considering the effective area available for the copolymer's deposition. In fact, the lower the nanoparticle concentration, the smaller the area available for the adsorption; therefore, it is expected that there will be stronger competition between the Pluronic F-127 molecules and micelles for occupying the nanoparticles' surface, resulting in the formation of more disordered layers with a higher fraction of Pluronic F-127 monomers protruding into the aqueous phase. This demonstrates the complexity of the concentration-dependent adsorption behavior and highlights the importance of understanding surface interactions. In a similar situation to the above, the effect of the silica nanoparticle concentration on the adsorption of Pluronic F-127 at a constant concentration (1 mg/mL) was examined. As the concentration of silica nanoparticles increased, there was a decrease in the apparent hydrodynamic diameter of the copolymer–nanoparticle complexes. The association of Pluronic F-127 and silica nanoparticles in the aqueous bulk can also be evaluated in terms of the effective charge of the decorated colloids, evaluated as the $\zeta$ potential (see Figure 3).

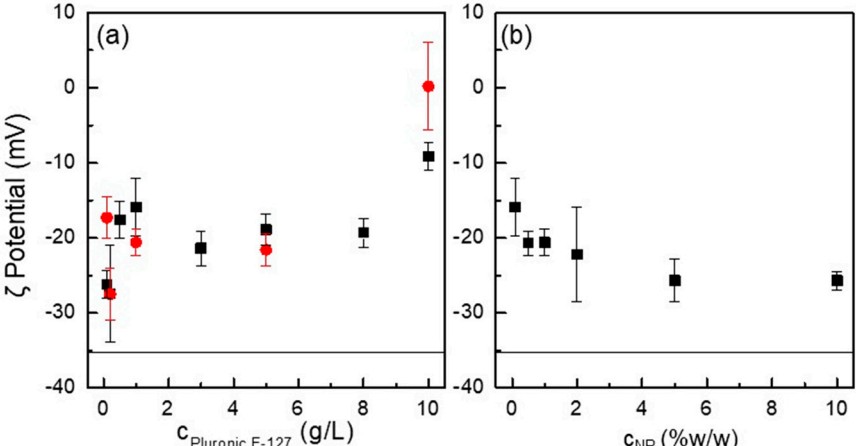

**Figure 3.** (**a**) Dependence of the $\zeta$ potential on the concentration of Pluronic F-127 for the mixed dispersions. (■) $c_{NP} = 0.1\%$ *w/w* and (●) $c_{NP} = 1\%$ *w/w*. (**b**) Dependence of the $\zeta$ potential on the nanoparticle concentration for silica nanoparticle dispersions with Pluronic F-127, where the concentration of Pluronic F-127 is fixed at 1 mg/mL. The lines in both panels indicate the $\zeta$ potential value corresponding to the bare silica nanoparticles, and the error bars represent the standard deviation of five independent measurements.

The results show a progressive shift of the $\zeta$ potential value, moving from that of the unmodified particles to values close to electroneutrality. Based on this result, it can be assumed that the formation of the Pluronic F-127 shell on the silica surface shields the negative charge on the nanoparticle. Consequently, the stability mechanism of the particles in the aqueous medium shifts from electrostatic stabilization to steric stabilization, leading to the formation of core–shell colloids. These colloids consist of an inner core composed of silica nanoparticles, while the outer shell is formed by the capping Pluronic F-127 layer (see references [17,18] for a sketch of the core–shell decorated colloids). In addition, the dependence of the coating density on the number of nanoparticles dispersed in the aqueous medium is supported by the observed reduction in the effective charge of the copolymer-decorated nanoparticles with increasing nanoparticle concentration. As mentioned earlier, a higher particle concentration results in a limited availability of Pluronic F-127 to coat each individual particle.

### 3.3. Adsorption of Pluronic-F-127-Decorated Silica Nanoparticles at the Water–Vapor Interface

The study of the adsorption of Pluronic-F-127-decorated silica nanoparticles at the water–vapor interface requires a careful investigation of the equilibrium interfacial tension of both the Pluronic-F-127- and particle-laden water–vapor interfaces. In addition, it is interesting to investigate the interfacial dilatational response of the formed layers once equilibrium is achieved. Understanding these aspects is of paramount importance when using interfacial layers to stabilize dispersed systems, and considering that in the system studied here, a water–vapor interface is considered. The information obtained may be relevant to improving the stability of foams [39–41].

#### 3.3.1. Interfacial Tension of the Water–Vapor Interface

Before discussing the adsorption of Pluronic F-127 and copolymer-decorated silica nanoparticles at the water–vapor interface, it is important to highlight that the bare silica nanoparticles exhibit minimal interfacial activity when considering their interaction with water–vapor interfaces. This observation is consistent with previous results reported in the literature [14,42,43]. The lack of significant interfacial activity may be attributed to reluctance of the nanoparticles to spontaneously adsorb at the water–vapor interface. However, it is also possible that they do adsorb to some extent but exert weak repulsive forces at the interfaces [44]. In contrast, the interfacial tension of the bare water–vapor interface is significantly affected by both Pluronic F-127 and copolymer-decorated nanoparticles. Figure 4 shows the relationship between Pluronic F-127 concentration and interfacial tension observed at the water–vapor interface after adsorption of either Pluronic F-127 or copolymer-coated nanoparticles.

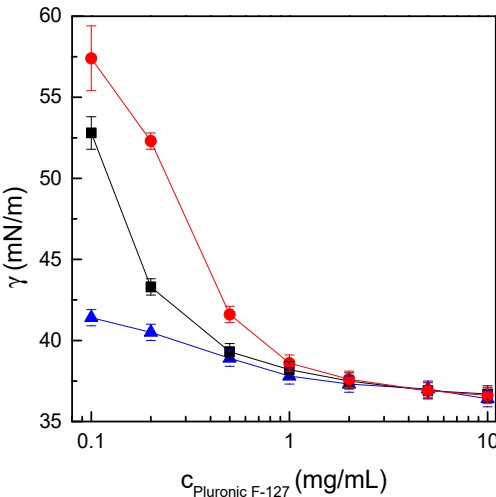

**Figure 4.** Pluronic F-127 concentration dependence of the interfacial tension of the bare water–vapor interface upon the adsorption of Pluronic F-127 or copolymer-decorated silica nanoparticles. (▲) $c_{NP}$ = 0% $w/w$ (Pluronic F-127 solutions), (■) $c_{NP}$ = 0.1% $w/w$ and (●) $c_{NP}$ = 1% $w/w$. The lines are guides for the eyes, and the error bars were obtained as the standard deviation of three replicates for each sample.

In the case of the interaction of Pluronic F-127 with the liquid–vapor interface, the behavior is consistent with that expected for the adsorption of an amphiphilic copolymer with surfactant characteristics. In fact, Pluronic F-127 is able to to reorganize at the interface with its most hydrophobic blocks, i.e., the poly(propylene oxide) ones, protruding towards the vapor phase, while the poly(ethylene oxide) blocks ensure the attachment of the copolymer to the liquid interface. Considering the investigated concentration range of Pluronic F-127, it is expected that the Pluronic F-127 is attached to the interface in a mushroom or brush conformation [45–48]. It is more interesting to evaluate the change in water–vapor interfacial tension with the Pluronic F-127 concentration when dispersions of

Pluronic-F-127-decorated silica nanoparticles are considered. The results show that for a fixed concentration of Pluronic F-127, the interfacial tension increases with the concentration of silica nanoparticles. This is similar to the effect observed by Ravera et al. [42] in their study of the adsorption of surfactant-decorated particles at the water–vapor interface. This type of phenomenon is different from that observed in other colloidal systems, such as polymer–surfactant mixtures, where the association process is accompanied by an enhanced decrease in the surface tension of the fluid–fluid interface [24].

For the case of Pluronic F-127 solutions in the investigated copolymer concentration range, the interfacial tension is on a plateau region related to the maximum coverage of the interface, or closer to such a situation (at the lowest Pluronic F-127 concentrations); for copolymer-decorated particles the results show higher values of interfacial tension than for the copolymer solutions at low Pluronic F-127 concentrations, and then as the Pluronic F-127 concentration increases, the interfacial tension decreases to values similar to those found for bare Pluronic F-127 solutions. These results can be explained by considering two different frameworks: The first assumes that the introduction of silica nanoparticles causes a depletion of Pluronic F-127 molecules from the aqueous phase to the surface of the silica nanoparticles, thereby reducing the effective concentration of free copolymer in the solution. This results in a situation characterized by the adsorption of the Pluronic F-127 in solution, leaving the copolymer-decorated particles dispersed in the aqueous phase. This reduction in the available concentration of Pluronic F-127 will result in higher interfacial tension values. The second scenario assumes that the adsorption of Pluronic F-127 onto the surface of the particles reduces their ability to reorganize at the interface and, thus, reduces the ability of the decorated particles to reduce the interfacial tension associated with the Pluronic F-127 shell compared to Pluronic F-127 solutions of the same concentration. Both scenarios are consistent with the dependence of the interfacial tension on the silica nanoparticle concentration. In fact, at low nanoparticle concentrations, the available area for Pluronic F-127 is small; therefore, both the depletion and the reduction in the degree of freedom of the attached Pluronic F-127 molecules will be very limited. Therefore, the effect of the particles on the interfacial tension of the water–vapor interface appears to be closer to that of bare Pluronic F-127 than when the particle concentration is increased. In the latter case, both the depletion and the reduction in the degree of freedom of Pluronic F-127 during adsorption will be greater. However, based on the results obtained for the bulk characterization of the dispersions, it can be assumed that the most important effect on the modification of the interfacial tension is related to the adsorption of the decorated particles at the water–vapor interface. Based on the above results, it is not expected that the formation of a densely packed particle-laden interface mediated by the adsorption of Pluronic F-127 on the particle surface can be responsible for the reduction in the interfacial tension. This is directly mediated by the Pluronic F-127 chains adsorbed on the particle surface, penetrating the interface and exerting long-range repulsions between the particles.

### 3.3.2. Dilational Elasticity of Particle-Laden Interfaces

The interfacial dilational elastic modulus, $\varepsilon'$, of the particle-laden interfaces was determined upon the equilibration of the adsorption process, i.e., when the interfacial tension reaches its equilibrium value. The frequency dependence of the interfacial dilational elastic modulus for Pluronic F-127 and copolymer-decorated silica nanoparticles with different concentrations of Pluronic F-127 is shown in Figure 5.

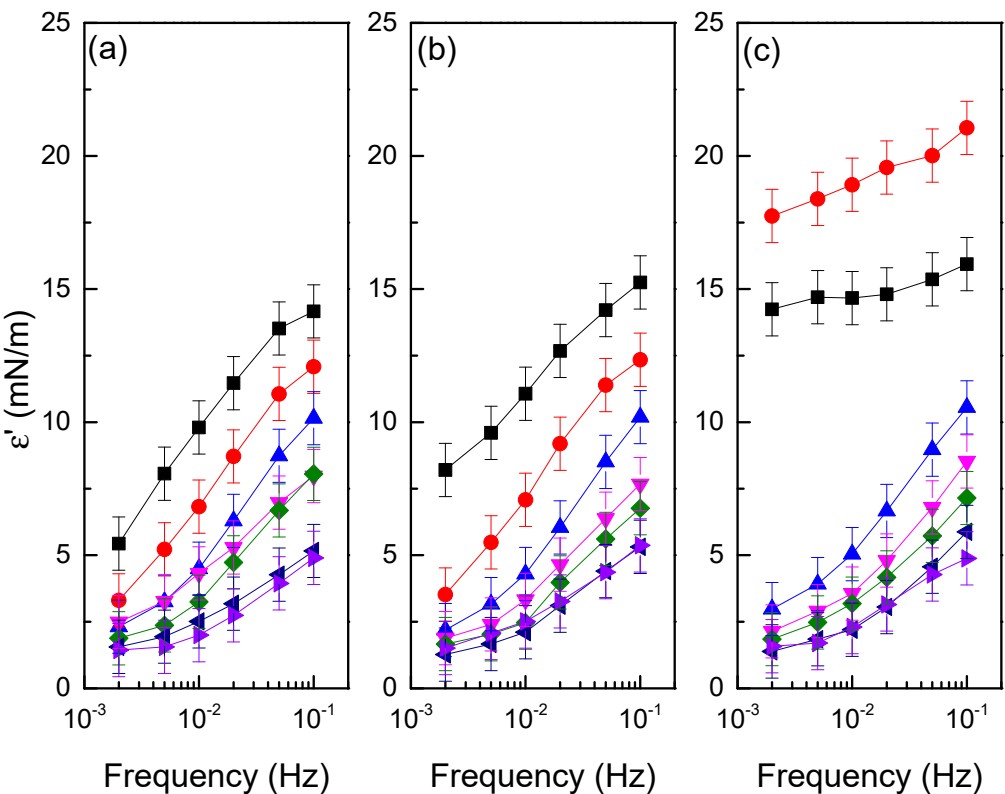

**Figure 5.** Dependence of the interfacial dilational elastic modulus on the frequency of the sinusoidal deformation (the amplitude of deformation was maintained at 10% for all of the experiments) for equilibrated layers of Pluronic F-127 and copolymer-decorated silica nanoparticles: (**a**) $c_{NP}$ = 0% *w/w* (Pluronic F-127 solutions), (**b**) $c_{NP}$ = 0.1% *w/w*, and (**c**) $c_{NP}$ = 1% *w/w*. The use of symbols with different colors indicates the different concentrations of Pluronic F-127 in the studied samples: (■) 0.1 mg/mL, (●) 0.2 mg/mL, (▲) 0.5 mg/mL, (▼) 1 mg/mL, (◆) 2 mg/mL, (◀) 5 mg/mL, and (▶) 10 mg/mL. The lines are guides for the eyes, and the error bars were obtained as the standard deviation of three replicates for each experiment.

As expected, in almost all of the experiments, as the deformation frequency increased, the value of the interfacial dilational elastic modulus also increased. However, for the lowest Pluronic F-127 concentrations and the highest particle concentrations, the interfacial dilational elastic modulus appeared to be weakly dependent on the frequency, which may indicate the formation of a solid-like particle-laden interface, in agreement with the discussion by Zhang et al. [49]. On the other hand, the copolymer concentration dependence of the interfacial dilational elastic modulus is not the same for all of the particle concentrations. At low particle concentrations, i.e., for layers formed only by Pluronic F-127 and layers formed when the particle concentration is 0.1% *w/w*, the interfacial dilational elastic modulus decreases with increasing concentration. This situation changes as the particle concentration increases (1% *w/w*), where ε′ initially increases with the Pluronic F-127 concentration and then, after reaching a threshold, the interfacial dilational elastic modulus begins to decrease with increasing concentration. The above behavior can easily be seen in the plot of the interfacial dilational modulus versus Pluronic F-127 concentration for experiments performed at a fixed frequency of $10^{-1}$ Hz (see Figure 6). It should be noted that the existence of a peak for the adsorption of colloids from dispersions where the particle concentration is 1% *w/w* has a real physical origin. In fact, considering the state of the monolayer at the lowest concentration range, an increase in the dilational elasticity with Pluronic F-127 concentration is consistent with the sharp decrease in surface tension observed in the isotherm shown in Figure 4. It is expected that a similar dependence may

appear for Pluronic F-127 solutions and dispersions with a particle concentration of 0.1% $w/w$ by decreasing the Pluronic F-127 concentration below the range studied here.

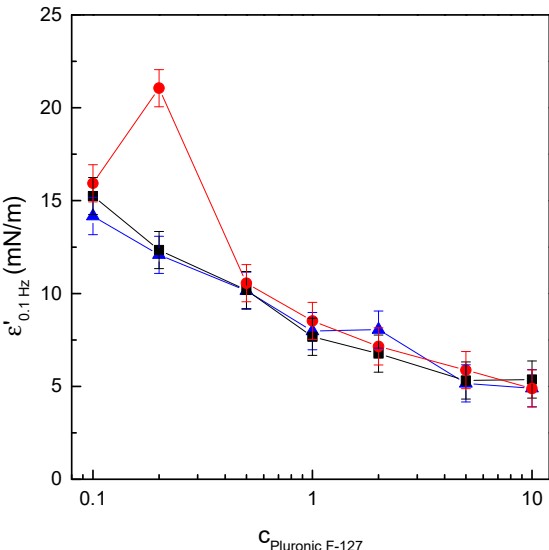

**Figure 6.** Dependence of the interfacial dilational elastic modulus on the Pluronic F-127 concentration for sinusoidal deformations (the amplitude of deformation was maintained at 10% for all of the experiments) with a fixed frequency of $10^{-1}$ Hz for equilibrated layers of Pluronic F-127 and copolymer-decorated silica nanoparticles. (▲) $c_{NP}$ = 0% $w/w$ (Pluronic F-127 solutions), (■) $c_{NP}$ = 0.1% $w/w$, and (●) $c_{NP}$ = 1% $w/w$. The lines are guides for the eyes, and the error bars were obtained as the standard deviation of three replicates for each experiment.

The analysis of the frequency dependence of the interfacial dilational elastic modulus shows the presence of a relaxation mechanism in the investigated frequency range, as evidenced by the inflection point in the curves. Considering the analyzed frequency range and its shift to higher values of frequency with the copolymer concentration, it can be assumed that the observed relaxation may correspond to the diffusion transfer of the copolymer (or copolymer-decorated particles) to the interface [10,42]. The results do not rule out the presence of additional relaxation processes occurring at the interface [10,50]. However, the limitation of the accessible frequency range analyzed by oscillatory barrier experiments prevents the evaluation of their role in the stabilization of the liquid–vapor interfaces.

The dependence of the interfacial dilational elastic modulus on the copolymer concentration, as shown in Figure 6, shows differences consistent with the different states of the interface according to the interfacial tension measurements. In fact, for Pluronic F-127 layers and those containing a low concentration of particles (0.1% $w/w$), the system is close to an interfacial tension plateau, i.e., the saturation of the interface, which explains the decrease in the dilatational modulus with the copolymer concentration. However, for the samples containing a particle concentration of 1% $w/w$, the interfacial tension isotherm for the lowest Pluronic F-127 concentration is in a region of steepest decrease, and this is consistent with the initial increase in the elastic modulus followed by a decrease when the interfacial state is close to the maximum adsorption, as evidenced by the interfacial tension plateau. In addition, the rheological measurements show values for the interfacial dilational elastic modulus that are very similar, shifted only by the presence of particles. In fact, this appears to be controlled solely by the presence of Pluronic F-127, again indicating the negligible role of the particles in controlling the interfacial properties of the interface. Therefore, the mechanical response of the interface suggests that the ability of Pluronic F-127 to reorganize at the water–vapor interface is essential in controlling the properties of the layers obtained, independent of the presence of particles. This can be understood by considering that the association of particles with Pluronic F-127 limits the ability of the latter to undergo its characteristic reorganizations at the interface, which is reasonable con-

sidering that some segments are attached to the surface of the particles and, therefore, limit the ability of Pluronic F-127 to reach its equilibrium conformation. Thus, considering the above scenario, the silica nanoparticles at the liquid–vapor interface act as obstacles to the reconfiguration of the copolymer chains. Once the Pluronic F-127 density is high enough to overcome the hindrance associated with the nanoparticles, the interfacial behavior of the decorated nanoparticles becomes reminiscent of what is expected for the bare copolymer.

## 4. Conclusions

The interaction between Pluronic F-127 and hydrophilic silica nanoparticles leads to the formation of stable complexes (copolymer-decorated nanoparticles) through hydrogen bonding between the oxyethylene groups of the copolymer and the non-dissociated silanol groups on the surface of the silica nanoparticles. The formation of these complexes favors the transport of the silica nanoparticles, which have poor interfacial activity, to the water–vapor interface and aids in their attachment.

The results obtained indicate that the adsorption of copolymer-decorated nanoparticles at the water–vapor interface does not lead to noticeable differences from the interfacial properties of Pluronic F-127 in the same concentration range, and only a shift in them is found, depending on the particle concentration. In fact, the density of the Pluronic F-127 shell determines the interfacial tension due to its effect on the reorganization of the Pluronic F-127 molecules at the interface. The same idea can explain the dependence of the interfacial dilational elastic modulus on the concentration of Pluronic F-127 and silica nanoparticles. In fact, particles at the liquid–vapor interface behave as simple obstacles, and when the concentration of Pluronic F-127 becomes high enough to overcome the hindrance induced by the particles, the interfacial behavior becomes reminiscent of that corresponding to bare Pluronic F-127 solutions. This means that the modification of particles with Pluronic F-127 opens interesting avenues for tuning the mechanical performance of liquid–vapor interfaces with a decisive impact on the interfacial stability, and for applications of particle-laden interfaces in the stabilization of dispersed systems. In particular, since this work focuses on water–vapor interfaces, the results can be used to understand the most fundamental physicochemical principles underlying the stabilization of particle-stabilized foams.

**Author Contributions:** Conceptualization, R.G.R. and E.G.; methodology, C.C. and A.R.-B.; software, C.C. and E.G.; validation, F.O., R.G.R. and E.G.; formal analysis, C.C. and E.G.; investigation, C.C., A.R.-B., F.O., R.G.R. and E.G.; resources, F.O. and R.G.R.; data curation, C.C. and E.G.; writing—original draft preparation, E.G.; writing—review and editing, C.C., F.O., R.G.R. and E.G.; visualization, C.C. and E.G.; supervision, E.G.; project administration, E.G.; funding acquisition, F.O., R.G.R. and E.G. All authors have read and agreed to the published version of the manuscript.

**Funding:** This work was funded by MICIN (Spain) under grant PID2019-106557GB-C21, and by the E.U. within the framework of the European Innovative Training Network-Marie Skłodowska-Curie Action NanoPaInt (grant agreement 955612).

**Institutional Review Board Statement:** Not applicable.

**Informed Consent Statement:** Not applicable.

**Data Availability Statement:** Data are available upon reasonable request.

**Acknowledgments:** The Centro de Espectroscopía y Correlación of the Universidad Complutense de Madrid is acknowledged for the use of their facilities.

**Conflicts of Interest:** The authors declare no conflict of interest. The funders had no role in the design of the study; in the collection, analyses, or interpretation of data; in the writing of the manuscript; or in the decision to publish the results.

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
