# Peer review of "Adsorption of Mixed Dispersions of Silica Nanoparticles and an Amphiphilic Triblock Copolymer at the Water–Vapor Interface"

_applsci, doi:10.3390/app131810093_

Round 1

Reviewer 1 Report

The MS represents non-covalent modification of bare negatively charged silica nanoparticles by the well-known triblock copolymer F-127. It is well-known that the adsorption of F-127 is greatly enhanced if the silica surface is decorated by amino-groups or in the case of the presence of the surface exposed metal centers. Nevertheless, the presence of non-dissociated Si-OH is enough to favor the adsorption. This fact has been already documented in literature. The new result of the present work is the estimation of the fact that SNs are incorporated into the water/air interfacial layer. Authors use DLS, surface tension and the interfacial dilational elastic modulus techniques to prove the incorporation. Indeed, only the latter technique give more or less reliable evidence of the involvement of the SNs in the interfacial layer. The mild acidifying can induce the adsorption of F-127 onto SNs due to the growing extent of Si-OH groups. May be a decrease in pH may favor the binding of F-127 with SNs, which can increase their incorporation into the interfacial layer? The MS should be significantly shortened, some schemes should be added to make the work more sound. 

Author Response

The MS represents non-covalent modification of bare negatively charged silica nanoparticles by the well-known triblock copolymer F-127. It is well-known that the adsorption of F-127 is greatly enhanced if the silica surface is decorated by amino-groups or in the case of the presence of the surface exposed metal centers. Nevertheless, the presence of non-dissociated Si-OH is enough to favor the adsorption. This fact has been already documented in literature. The new result of the present work is the estimation of the fact that SNs are incorporated into the water/air interfacial layer. Authors use DLS, surface tension and the interfacial dilational elastic modulus techniques to prove the incorporation. Indeed, only the latter technique give more or less reliable evidence of the involvement of the SNs in the interfacial layer. The mild acidifying can induce the adsorption of F-127 onto SNs due to the growing extent of Si-OH groups. May be a decrease in pH may favor the binding of F-127 with SNs, which can increase their incorporation into the interfacial layer? The MS should be significantly shortened, some schemes should be added to make the work more sound. 

We agree with the reviewer that the reduction of the effective charge of the silicon dioxide nanoparticles may increase the number of available silanol, which may increase the effectiveness of the Pluronic F-127 adsorption. However, the reduction of the effective charge of the silicon dioxide nanoparticles may induce a destabilization of the silicon dioxide nanoparticles increasing their aggregation. Therefore, a mild acidification of the dispersion may lead to a competition between the adsorption of the Pluronic F-127 on the nanoparticles and the aggregation of the latter. This may introduce some difficulties in the evaluation of the bulk properties of the mixed dispersion, and for sure would modify the interfacial properties of the system. We have introduced a brief statement on the text for explaining this situation.

We have shortened the text when it was possible without compromising the content. However, in many points the shortening is not possible without affecting to the information provided by the work. Moreover, the substitution of content by schemes goes against the clarity of the manuscript, and therefore we have preferred to avoid any additional scheme in the text.

We thank to the reviewers for the comments, they were very useful for improving the quality of our manuscript.

Reviewer 2 Report

I have carefully read the article entitled “adsorption of Mixed Dispersions of Silica Nanoparticles and an Amphiphilic Triblock Copolymer at the Water/Vapor Interface” and find the work relevant is fitting the scope of the journal. Went through the manuscript in detail. However, needs major revisions to be considered as publishable in the journal.

 Comments:

1.     English needs to be polished and grammar improved considerably throughout the whole manuscript.

2.     Abstract should be more specific and mention what was the achievement.

3.     Authors should precise the introduction part, generally, the introduction section should tell the readers why use this work and the benefits gain behind it. 

4.     Authors should establish a comparison of their results with that in the literature to clarify the novelty of the work.

Moderate editing of English language required

Author Response

I have carefully read the article entitled “adsorption of Mixed Dispersions of Silica Nanoparticles and an Amphiphilic Triblock Copolymer at the Water/Vapor Interface” and find the work relevant is fitting the scope of the journal. Went through the manuscript in detail. However, needs major revisions to be considered as publishable in the journal.

Comments:

  1. English needs to be polished and grammar improved considerably throughout the whole manuscript.

We have checked the grammar and spelling within the whole manuscript.

  1. Abstract should be more specific and mention what was the achievement.

We have modified the abstract according to the reviewer comment.

  1. Authors should precise the introduction part, generally, the introduction section should tell the readers why use this work and the benefits gain behind it.

We have modified the introduction according to the reviewer comment.

  1. Authors should establish a comparison of their results with that in the literature to clarify the novelty of the work.

We have introduced some comparison with the literature when it is possible.

We thank to the reviewers for the comments, they were very useful for improving the quality of our manuscript.

Reviewer 3 Report

The paper " Adsorption of Mixed Dispersions of Silica Nanoparticles and 2 an Amphiphilic Triblock Copolymer at the Water/Vapor 3 Interface " is an original research work describing the effect of surface modification of silica nanoparticles with an amphiphilic triblock copolymer (Pluronic F-127) on interfacial properties.  My recommendation is that the paper be published in Applied Sciences after minor corrections which I describe below:

1. The behavior of the e' vs. frequency curve for the sample with 0.2 mg/mL of Pluronic F-127 and a nanoparticle concentration of 1% w/w shown in Figure 5 (c) is the only one that deviates from the observed behavior in the rest of the samples (this can also be observed in Figure 6). The authors should verify that this behavior is not due to any procedural error during sample preparation (since the dilational rheometry test was conducted in triplicate). Consequently, the discussion of this section (page 10, lines 383 to 394) should be reviewed. The behavior observed in Figure 6 suggests that there is no dependency of e' on the nanoparticle concentration, considering the experimental error.

Author Response

The paper " Adsorption of Mixed Dispersions of Silica Nanoparticles and 2 an Amphiphilic Triblock Copolymer at the Water/Vapor 3 Interface " is an original research work describing the effect of surface modification of silica nanoparticles with an amphiphilic triblock copolymer (Pluronic F-127) on interfacial properties.  My recommendation is that the paper be published in Applied Sciences after minor corrections which I describe below:

  1. The behavior of the e' vs. frequency curve for the sample with 0.2 mg/mL of Pluronic F-127 and a nanoparticle concentration of 1% w/w shown in Figure 5 (c) is the only one that deviates from the observed behavior in the rest of the samples (this can also be observed in Figure 6). The authors should verify that this behavior is not due to any procedural error during sample preparation (since the dilational rheometry test was conducted in triplicate). Consequently, the discussion of this section (page 10, lines 383 to 394) should be reviewed. The behavior observed in Figure 6 suggests that there is no dependency of e' on the nanoparticle concentration, considering the experimental error.

We have checked the experiments, and we are sure that the result is not any procedural error. To ensure the reliability of our results, each measurement was performed with an independent monolayer, and in all the cases the results fall within the error bars. In addition, the result is the expected considering the state of the monolayer at such concentration. According to the surface tension isotherm, at the lowest concentrations, for the nanoparticle concentration of 1%w/w, the surface tension is in a region characterized by a sharp decrease, this must lead to the increase of the dilational elasticity to reach a maximum at a concentration equivalent to the saturation point of the adsorption. This is the situation that we found in our results, it is possible that if the concentration of Pluronic F-127 is decreased similar dependences would be found for bare copolymer and for the mixtures with 0.1%w/w of silica nanoparticles.

We have added an explanation to the text.

We thank to the reviewers for the comments, they were very useful for improving the quality of our manuscript.

Round 2

Reviewer 1 Report

The revised version is significantly improved. Indeed, the pH-effect should improve the deposition of F-127 in very narrow pH-range, since the acidification can trigger the aggregation process. I can understand the reluctance of the authors to do an additional experimental work, but pH-value is of key importance in such process. So, please introduce the applied pH-values in the Figure and Table Captions. This will make your work reproducible, and more interesting for the readers.

Minor editing if any.

Author Response

We thank to the reviewer for the comment, instead of including the pH in all the legends, we have introduced a comment in the methodological section where the pH value is stated. Thus, we avoid unneccesary repetition in the manuscript.

We have checked the grammar and spelling within the manuscript.

Reviewer 2 Report

The manuscript was improved and it can be accepted for publication in current form 

Minor editing of English language required

Author Response

We have checked grammar and spelling within the manuscript.